# The correlation between Arctic sea ice, cloud phase and radiation using A-train satellites

Grégory V. Cesana[1,2], Olivia Pierpaoli[3,2], Matteo Ottaviani[4,2], Linh Vu[3,2], Zhonghai Jin[2], and Israel Silber[5,6]

[1]Center for Climate Systems Research, Columbia University, New York, NY
[2]NASA Goddard Institute for Space Studies, New York, NY
[3]Department of Atmospheric Sciences, University of Washington, Seattle, WA
[4]Terra Research Inc, Hoboken, NJ 07030
[5]Department of Meteorology and Atmospheric Science, Pennsylvania State University, University Park, PA, USA
[6]now at Atmospheric, Climate, and Earth Sciences Division, Pacific Northwest National Laboratory, Richland, WA, USA

**Correspondence:** Grégory V. Cesana (gregory.cesana@columbia.edu)

**Abstract.** Climate warming has a stronger impact on Arctic climate and sea ice cover (SIC) decline than previously thought. Better understanding and characterizing the relationship between sea ice, clouds and the implications for surface radiation is key to improving our confidence in Arctic climate projections. Here we analyze the relationship between sea ice, cloud phase and surface radiation over the Arctic, defined as north of 60˚N, using active- and passive-sensor satellite observations from three
different datasets. We find that all datasets agree on the climatology and seasonal variability of total and liquid-bearing (liquid and mixed-phase) cloud covers. Similarly, our results show a robust relationship between decreased SIC and increased liquid-bearing clouds in the lowest levels (below 3 km) for all seasons (strongest in winter) but summer, while increased SIC and ice clouds are positively correlated in two of the three datasets. A refined map correlation analysis indicates that the relationship between SIC and liquid-bearing clouds can change sign over the Bering, Barent and Laptev seas, likely because of intrusions
of warm air from low latitudes during winter and spring. Finally, the increase of liquid clouds resulting from decreasing SIC is associated with enhanced radiative cooling at the surface. Our findings indicate that the newly formed liquid clouds reflect more shortwave (SW) radiation back to space compared to the surface, generating a cooling effect of the surface, while their downward longwave (LW) radiation is similar to the upward LW surface emission, which has a negligible radiative impact on the surface. This overall cooling effect should contribute to dampening future Arctic surface warming as SIC continues to
decline.

## 1   Introduction

Clouds can have different radiative effects over the Arctic (Curry et al. (1996); Shupe and Intrieri (2004)), defined as north of 60˚N: cooling in summer generated by sunlight reflection and warming in winter through LW radiative heating of the surface. In a region where the warming can be up to four times larger than in the rest of the world (Boeke and Taylor (2018)), it is crucial
to determine how clouds respond to climate change and whether their feedback will enhance or dampen the warming. While cloud feedback was shown to play a minor role in polar amplification in CMIP5 (Pithan and Mauritsen (2014), Middlemas

et al. (2020)), studies using the most recent climate model generation (CMIP6) unveil a strong impact of extratropical mixed-phase clouds on global climate (Zelinka et al. (2020)). Part of the uncertainty in determining cloud variability in polar regions stems from the extent to which Arctic clouds respond to sea ice loss (Kay and Gettelman (2009); Morrison et al. (2018); Taylor and Monroe (2023)), which is still debated. Within the context of substantial decrease of sea ice extent over the past few decades (Kim et al., 2023), determining how clouds may be affected by this tremendous change is necessary to simulate realistic projections of polar climate.

The analysis of various observational datasets has led to consensus on the sensitivity of cloud fraction to sea-ice extent variability in the fall but not in the summer (e.g., (Kay and Gettelman (2009); Morrison et al. (2018)). As the sea-ice cover (SIC) decreases in the fall, the cloud fraction increases, mostly attributable to low-level liquid clouds (Morrison et al. (2018)). A more recent study found a larger cloud fraction over open ocean than over sea ice also in winter and spring (Taylor and Monroe (2023)). In general, these studies try to better understand the relationship between sea ice and clouds at the process level and show that clouds are most likely controlled by air-surface coupling. However, they focus on one specific dataset over a limited amount of years, and disregard the response of ice clouds and more broadly free-troposphere clouds, which can have a notable radiative impact (L'Ecuyer et al. (2019)).

In this study, we build on their results to analyze the relationship between sea ice and clouds on longer time scales to advance our understanding of Arctic climate and with the hope to provide constraint for climate models. We use cloud observations from three different datasets that discriminate between liquid, ice and mixed-phase clouds. The first dataset is constrained to years 2007-2010 and is based on CloudSat-CALIPSO retrievals (DARDAR). The second is a new dataset based on CALIPSO-GOCCP that documents all liquid, mixed-phase and ice-only clouds from 2007 to 2020. The third is a passive-sensor satellite dataset based on MODIS retrievals (CERES) and also provides surface flux retrievals. We also analyze all seasons over the whole Arctic Ocean. After describing the datasets, we show their climatology of the different Arctic cloud types for all seasons. We then perform a correlation analysis of the SIC with all cloud types. Additionally, we focus on the vertical structure of these cloud types as a function of sea-ice conditions. Finally, we investigate the radiative impact of these clouds at the surface.

## 2 Datasets

### 2.1 CALIPSO-PHACT

The PHAse Cloud Type (PHACT) product development has been guided by recent ground-based observations to document the topmost liquid-bearing cloud layers for different cloud types (ice over liquid, liquid only, liquid seeded by ice above, mixed-phase, multilayer and single layer etc.) and for thin (optical depth $\lesssim 3$) and opaque (optical depth $\gtrsim 3$) clouds. PHACT, which will be fully described in a separate paper, uses instantaneous CALIPSO-GOCCP profiles (Cesana and Chepfer (2013)); Chepfer et al. (2010)). These profiles document cloud properties obtained from near-nadir lidar profiles at a 333 m along-track resolution and for 480 m altitude bins. Cloud phase diagnostics are based on the cloud particle sphericity instead of temperature, in contrast with many passive sensors. Liquid- and ice-dominated altitude bins are discriminated using the polarization state of the laser return, which changes when backscattered by a non-spherical crystal as opposed to spherical droplets. In highly

reflective layers, the distinction between the two water phases is more ambiguous, because of multiple scattering and noise, and results in undefined-phase clouds, which often correspond to mixed-phase clouds (Cesana et al. (2016)).

PHACT 2D cloud covers and 3D cloud fractions differ from CALIPSO-GOCCP cloud phase statistics in two main ways. First, PHACT provides ice-only (no liquid in the column) and mixed-phase (liquid with ice below, contiguous or not) cloud covers, in addition to the traditional liquid (with possibly ice above) phase category. Second, the phase cloud cover is computed as the number of ice, liquid or mixed-phase cloudy profiles divided by the total number of profiles, consistent with the common definition of cloud cover:

$$CC_{phase} = N_{phase}/N_{profiles} \tag{1}$$

Where phase means either ice, liquid or mixed-phase.

As a result, the sum of ice-only, liquid, and mixed-phase cloud covers cannot be greater than 1. We note a large difference in the magnitude of the vertical cloud profiles between CALIPSO-GOCCP and PHACT because PHACT is focused on the topmost liquid-layer whereas CALIPSO-GOCCP documents all available vertical levels. Additionally, PHACT contains a mixed-phase category which is not available in CALIPSO-GOCCP. Mixed-phase clouds are diagnosed when ice or undefined-phase clouds are retrieved below liquid, either directly underneath or not. Undefined-phase clouds are clouds that are located underneath highly reflective clouds (i.e., optically thicker, Cesana and Chepfer (2013)) and have been shown to be most likely mixed-phase clouds at subfreezing temperatures (Cesana et al. (2016)). A validation study against in situ aircraft measurements shows that the maximum disagreement fraction between CALIPSO-GOCCP instantaneous profiles – used in PHACT – and five in situ aircraft flights is $\sim 11.8$ % when accounting for in situ aircraft measurement uncertainty (Cesana et al. (2016); their Table 3).

## 2.2 CloudSat-CALIPSO DARDAR

The DARDAR product (liDAR/raDAR; Delanoë and Hogan (2010)) uses both CALIPSO and CloudSat observations to retrieve vertical profiles of cloud properties with a vertical and horizontal resolution of 60 m and 1.7 km, respectively. The DARDAR cloud phase algorithm is based on the complementarity of the high sensitivity of the 532 nm lidar to small and spherical liquid droplets and the 94 GHz radar to large ice crystals. In addition, the algorithm also uses cloudy layer temperature and cloud geometrical thickness to determine cloud phase. DARDAR distinguishes between ice, mixed, supercooled and warm liquid clouds. A more detailed explanation of the algorithm is given by Mioche et al. (2015). Since DARDAR is not provided as a gridded Level3 product (as opposed to CALIPSO-GOCCP, CALIPSO-PHACT and CERES), we have processed the vertical profiles for the available time period, computed the cloud phase covers following the same definition as CALIPSO-PHACT and computed pan-Arctic statistics onto a 1˚x1˚ grid, which was then interpolated to 2.5°x2.5° grid. Similarly, we have used a cloud phase classification for the 2D maps so that the sum of ice-only, liquid and mixed-phase cloud covers cannot be greater than 1 and, hence, is equal to the total cloud cover. Should we have used ice-containing and liquid-containing 2D cloud covers we could have accounted for liquid or ice two times in the same profile, hence a cloud cover greater than 1.

We are not aware of a formal evaluation of DARDAR product but we note that when the lidar signal is fully attenuated (i.e., no more signal), the DARDAR algorithm attributes the ice phase to any cloudy pixels at subfreezing temperatures, which may lead to an overestimate of the ice cloud covers. In addition, Mioche et al. (2015) mention a potential overestimate of mixed-phase cloud due to excessive supercooled liquid detections, which may have been mitigated in the newer version of DARDAR product, which is used here.

## 2.3 CERES

To study the surface fluxes, we use Clouds and the Earth's Radiant Energy System (CERES) FluxByCldTyp – Level 3 (Sun et al. (2022)), which also contains information about the cloud phase. The computation of all-sky and clear-sky fluxes assumes the same surface type, which means that the surface albedo is accounted for in the computations of cloud radiative effect (CRE) values (Section 3). The cloud and cloud phase information is based on MODIS cloud properties. The phase is retrieved for daytime only using the newest MODIS collection 6 (MODIS-C6) cloud phase algorithm, which employed a decision tree logic based on four independent tests (Marchant et al. (2016)): cloud top temperature, tri-spectral infrared test using difference in brightness temperatures, a $1.38\mu$m test to determine the presence of cirrus clouds and a tri-spectral cloud effective radius test. We note that MODIS-C6 cloud phase has a large agreement fraction with CALIPSO science team cloud phase retrievals (up to 90%, Marchant et al. (2016)) – for clouds that are detected by both instruments – and that the use of daytime-only observations is a limiting factor in our analysis (no cloud phase data during the Arctic winter). The original grid of CERES dataset is 1˚x1˚.

We use data from the 2007-2020 overlapping period for PHACT and CERES. DARDAR data is available for 2007-2010 and 2013-2017, however, this latter period is limited to daytime observations, and using it could introduce biases in the comparison. Although DARDAR 2013-2017 and 2007-2010 shows differences in the total cloud cover that are comparable to that of PHACT over the same time periods (Fig. S1 and S2, bottom right plots), major differences appear in DARDAR cloud phase partitioning that do not in PHACT (Fig. S1 and S2, bottom rows). These differences suggest that using daytime-only data has a strong impact on DARDAR phase diagnostics.

Finally, all datasets are projected onto a 2.5˚x2.5˚ grid. We note that we find no significant impact of the spatial resolution – i.e., using either 2.5˚x2.5˚ and 1˚x1˚ grid – for PHACT total, ice and liquid cloud covers over the Arctic (not shown).

## 3 Results

Figure 1 shows a comparison of ice, liquid, mixed-phase and total cloud covers for all the datasets. All three datasets agree very well on the total cloud cover Interestingly, the datasets also detect a very similar liquid cloud cover (in terms of means and pattern correlations, Table 1 and Fig. S3) with a sharp contrast between land and sea and maxima over the Laptev, Barent and Greenland seas, even though they use independent methods to retrieve liquid layers. We note that CERES exhibits the largest amount of liquid clouds for two reasons (47%). First, it does not distinguish between mixed-phase and liquid clouds and second, it doesn't include the winter season north of 70˚N, which is the season with the least liquid cloud amount (Cesana et al. (2012), their Fig. 1). When accounting for all liquid-bearing clouds (i.e., liquid-only and mixed-phase clouds), the differences

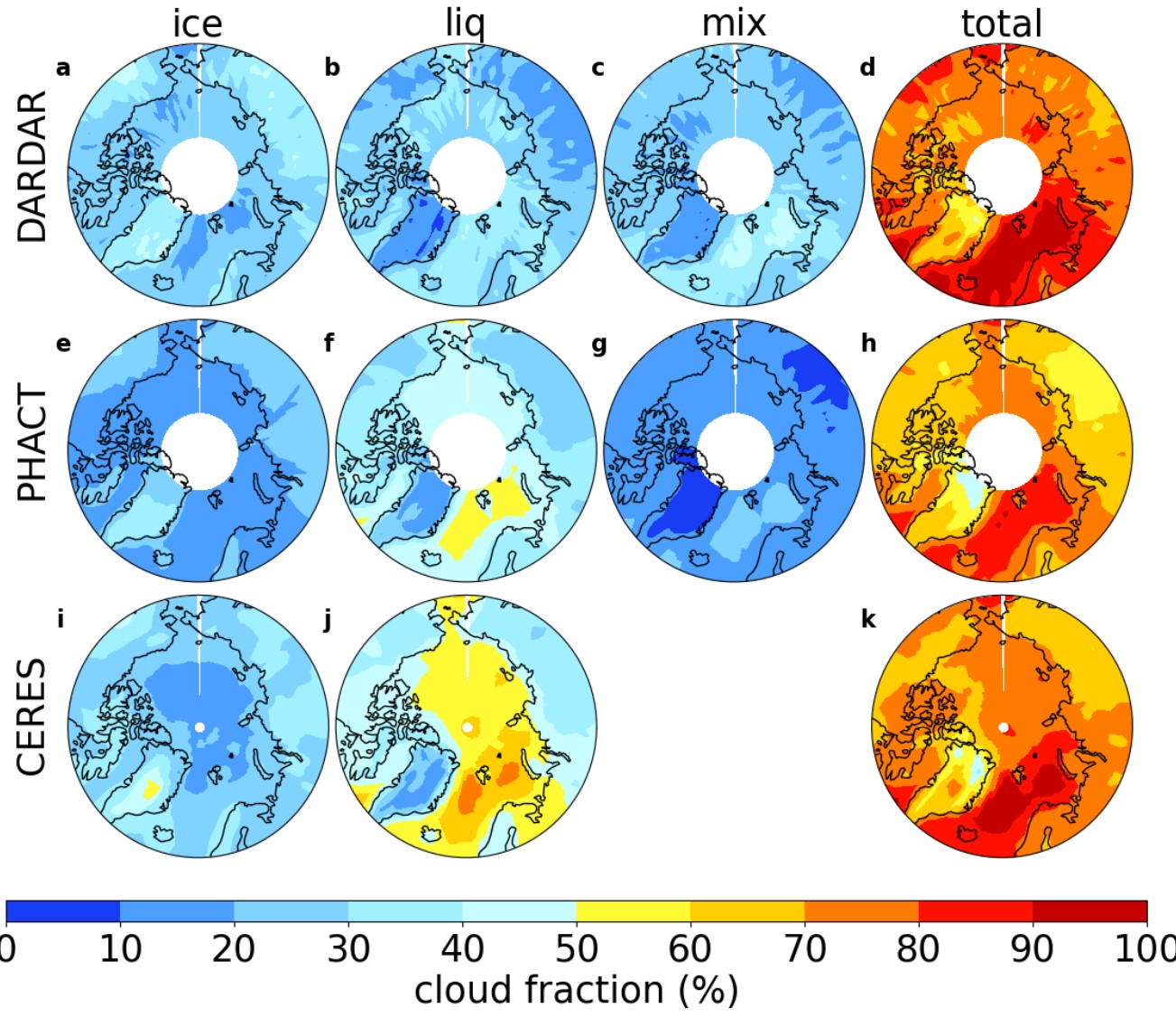

**Figure 1.** Average cloud ice, liquid, mixed-phase and total cloud covers (%) for DARDAR (2007-2010; Delanoë and Hogan (2010)), PHACT (2007-2020) and CERES (2007-2020; Sun et al. (2022)).

**Table 1.** Arctic area-weighted mean cloud covers (%) for all datasets from 2007-2010. We note a robust consistency in the liquid-bearing cloud cover among all datasets.

| (%) | DARDAR | PHACT | CERES ($< 82°$N) | CERES |
|---|---|---|---|---|
| **Total** | 77 | 68 | 73 | 73 |
| **Ice** | 27 | 18 | 29 | 29 |
| **Liquid** | 25 | 36 | 44 | 44 |
| **Mixed** | 25 | 14 | N/A | N/A |
| **Liquid-bearing** | 51 | 50 | 44 | 44 |

among datasets drastically lessens: 51%, 50% and 44% for DARDAR, PHACT and CERES, respectively. It is worth noting that although CERES observations come from a passive sensor instrument, which makes it more challenging to detect clouds and retrieve their properties over bright surfaces, they are in good agreement with the other two active-sensor observations. The mixed-phase cloud category is also consistent between DARDAR and PHACT (pattern correlation r = 0.56). Yet DARDAR diagnoses slightly more mixed-phase clouds than PHACT (25% compared to 14%), likely because the radar is able to penetrate the clouds deeper and detect precipitating ice below. In this regard, PHACT can be viewed as a lower bound in terms of mixed-phase cloud cover. On average, these mixed-phase clouds account for about 27% of liquid-bearing clouds in PHACT and 50% in DARDAR (Fig. S4, Tab. S1), which is closer to 73% from Arctic ground-based measurements using a radar with a cloud detection sensibility similar to CloudSat (Silber et al. (2021)). These discrepancies are attributable to lidar attenuation in the case of PHACT, which does not allow the lidar beam to reach the mixed-phase layer, to ground clutter in the case of CloudSat-based measurements, which prevents retrieving hydrometeors below 500 m, and to spatial variability (i.e., one data point in ground-based measurements against an Arctic-wide average in spaceborne measurements). Finally, the greatest differences between the datasets in terms of pattern correlation and cloud covers come from the ice clouds. DARDAR and CERES show more ice clouds than PHACT, and their pattern correlation are smaller than those from liquid clouds. To diagnose cloud cover, the DARDAR algorithm uses CALIPSO level2 product, which averages the lidar signal along track up to 80 km to be able to retrieve the thinnest cirrus clouds. This may explain why DARDAR detects more ice clouds than PHACT – although they both use CALIPSO observations but may also cause false positive detections (Cesana et al. (2016)). CERES reports larger ice cloud cover than PHACT because it does not diagnose ice-only cloud column as in PHACT. When accounting for ice above liquid clouds as well as ice-only clouds, consistent with CERES observations, PHACT detects more ice clouds than CERES (Fig. S5).

Figure 2 shows the seasonal variability of each cloud type over the Arctic for the three datasets. All datasets exhibit a maximum in Fall and a minimum in Winter in the total cloud cover, mostly driven by liquid-containing clouds (liquid and mixed-phase), which is consistent with previous findings (e.g., Cesana et al. (2012); Lacour et al. (2017); McIlhattan et al. (2017); Mioche et al. (2015); Shupe (2011)). That seasonality of liquid clouds is mainly attributable to environmental conditions

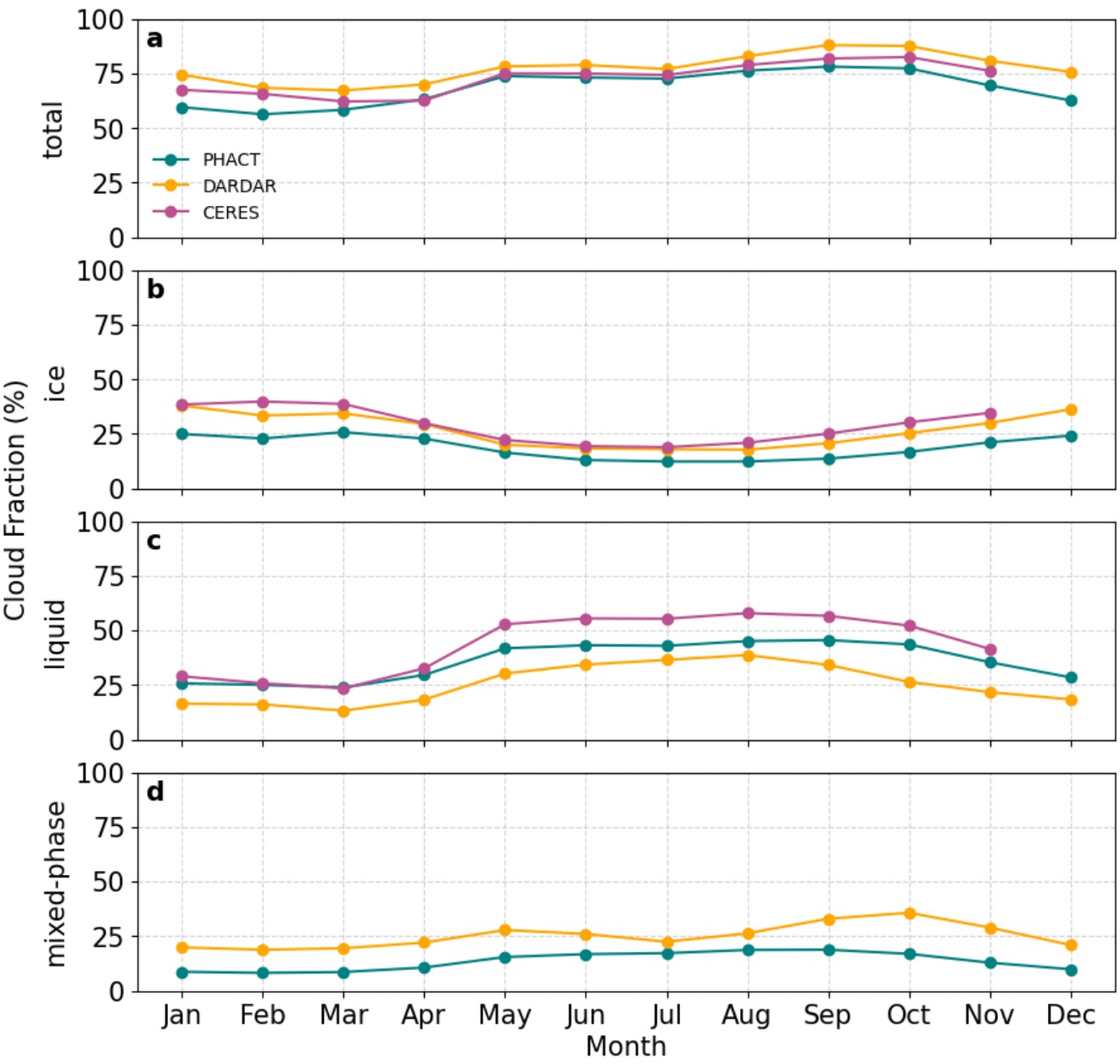

**Figure 2.** Seasonal variability of the Arctic cloud cover (%) for DARDAR (orange, 2007-2010), CERES (purple, 2007-2020) and PHACT (green, 2007-2020).

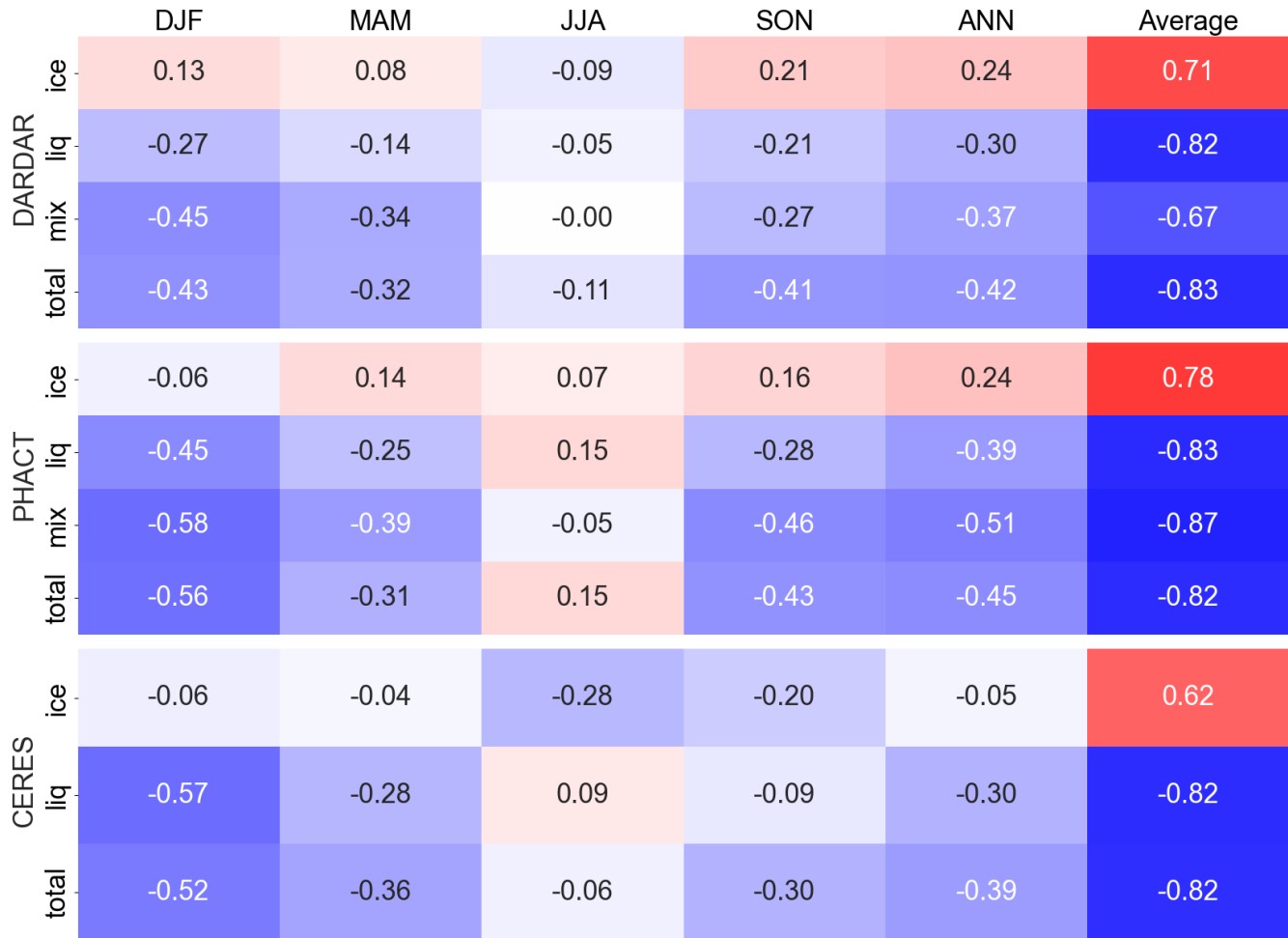

|  |  | DJF | MAM | JJA | SON | ANN | Average |
|---|---|---|---|---|---|---|---|
| **DARDAR** | ice | 0.13 | 0.08 | -0.09 | 0.21 | 0.24 | 0.71 |
|  | liq | -0.27 | -0.14 | -0.05 | -0.21 | -0.30 | -0.82 |
|  | mix | -0.45 | -0.34 | -0.00 | -0.27 | -0.37 | -0.67 |
|  | total | -0.43 | -0.32 | -0.11 | -0.41 | -0.42 | -0.83 |
| **PHACT** | ice | -0.06 | 0.14 | 0.07 | 0.16 | 0.24 | 0.78 |
|  | liq | -0.45 | -0.25 | 0.15 | -0.28 | -0.39 | -0.83 |
|  | mix | -0.58 | -0.39 | -0.05 | -0.46 | -0.51 | -0.87 |
|  | total | -0.56 | -0.31 | 0.15 | -0.43 | -0.45 | -0.82 |
| **CERES** | ice | -0.06 | -0.04 | -0.28 | -0.20 | -0.05 | 0.62 |
|  | liq | -0.57 | -0.28 | 0.09 | -0.09 | -0.30 | -0.82 |
|  | total | -0.52 | -0.36 | -0.06 | -0.30 | -0.39 | -0.82 |

**Figure 3.** Colored table of spatial (first five columns) and temporal (last column) correlations between cloud types and SIC for all seasons and for DARDAR (2007-2010), PHACT (2007-2020) and CERES (2007-2020). Dark red and blue colors indicate strong correlations and anti-correlations, respectively.

that favor liquid cloud formation, i.e., more moisture and mild temperature in summer and transition seasons as opposed to winter. Yet we note some small differences between CERES and the CALIPSO-based observations during the winter months, most likely because of the use of daytime only observations in CERES, which restricts the latitudes during these months. As expected, ice cloud cover reaches its lowest value in summer (July) in all datasets, while its maximum occurs in the winter. Using 2007-2010 instead of 2007-2020 does not change qualitatively or quantitatively our results (Fig. S6).

Next, we explore the relationship between clouds and SIC variability. Given the relatively good agreement in the representation of the seasonal variability of liquid and total cloud covers between all datasets, we expect to find robust relationships between these clouds and SIC.

We compute spatial correlations between SIC and cloud covers from each cloud type for our three cloud datasets. Our SIC comes from ERA5 reanalysis, but we note that no substantial differences are found using a different dataset (Hadley Centre Sea Ice and Sea Surface Temperature, not shown). Furthermore, we focus on those grid boxes in which SIC varies over the time period, that is averaged SIC is greater than 0.01 (excluding open-ocean only grid boxes) and smaller than 0.99 (excluding sea-ice-covered only grid boxes). Using a more restrictive thresholds (0.02 and 0.98 or 0.03 and 0.97) does not affect our results qualitatively (Table S1). Our results are consistent with previous literature. We find large and significant negative correlations between SIC and liquid and total cloud covers from all datasets (Figure 3). In addition, our results show little to no correlation between SIC and liquid cloud cover for the summer, consistent with previous studies (Kay and Gettelman (2009); Morrison et al. (2018)). More surprisingly, our analysis indicates that the ice cloud cover is somewhat positively correlated with SIC in all datasets, though less so in CERES likely because of liquid cloud contamination, which, to our knowledge, has not been reported in the literature before. It is also interesting to note that these correlations are stronger when comparing temporal variability of SIC and cloud covers (temporal correlation) although slightly weaker when using all grid boxes (Figures 4 5 6) rather than first averaging across the Arctic (Figure 2 last column). This discrepancy might be indicative of the influence of local parameters other than sea ice on clouds; when looking at a pan-Arctic perspective, this local influence fades out and the presence of sea ice is associated with synoptical meteorological conditions that favor ice cloud formation and hinders that of liquid clouds.

The correlation maps help us better understand the variability of the relationships between each cloud type and SIC as a function of the seasons (Figures 4 5 6). For liquid clouds (second row), the correlations are generally negative in the fall and spring while in summer little correlation is found, and in winter, it is mostly negative except over the Bering, Barent and Laptev seas. This exception might be explained by the incursion of moist and warm air from the Aleutian low – a frequent large-scale atmospheric pattern during the Arctic cold season (Overland et al. (1998)). The advection of moist air strongly affects Arctic surface temperatures (Shulski et al. (2010)). This moist air incursion results in the formation of liquid clouds at subfreezing temperatures in that area, while the sea ice continues to build up, even though strong Aleutian lows can reduce sea-ice growth (Walsh et al. (2017), Dörr et al. (2021)). Since these correlations occur mostly over grid boxes in which SIC is close to 1 – and does not vary much – the total correlation (Figure 3) remains highly negative, even more so than in SON. We speculate that it is more difficult for liquid clouds to form over sea ice than it would be in fall because of the lower temperatures – for similar environmental conditions – hence the stronger negative correlations in winter. We find similar patterns for the mixed-phase and total cloud correlation maps, which are clearly dominated by the response of liquid-containing clouds (liquid-only and mixed-phase clouds). Furthermore, we find that the correlations are even stronger for mixed-phase clouds than for liquid clouds in PHACT. In this dataset, the mixed-phase cloud are more opaque than liquid clouds. Recent literature suggests that opaque clouds are at a more mature stage of their lifecycle than thin clouds (Silber et al. (2020)), and therefore, they would have to linger over open ocean for longer to reach that mature stage, which could explain the stronger correlations of mixed-phase clouds with open ocean compared to optically thinner liquid clouds. The results are more diverse among the datasets when it comes to ice clouds. The correlations are mostly positive in PHACT (except in winter). DARDAR correlations are consistent with those form PHACT in spring and fall while CERES correlations are both negative and positive depending on

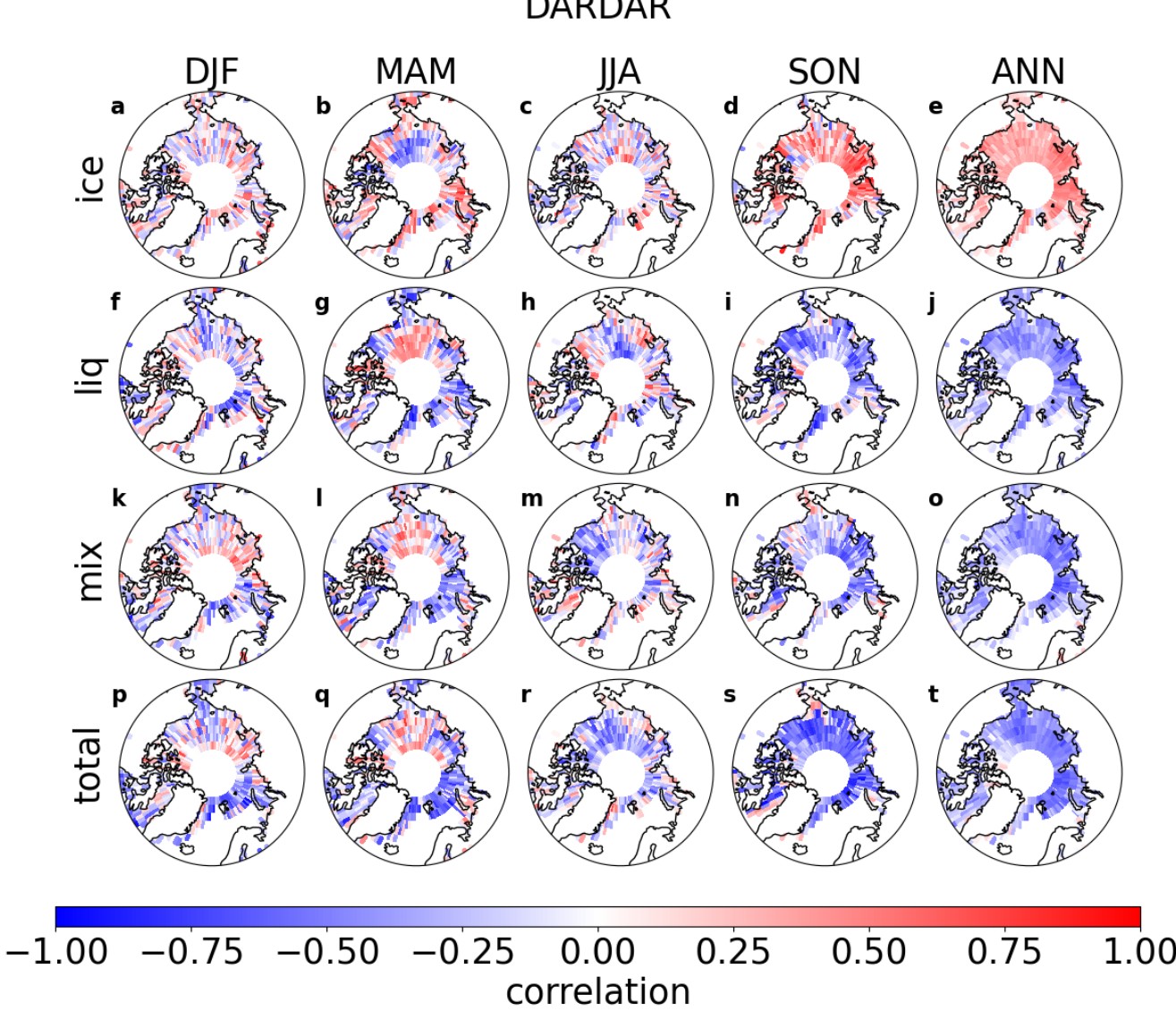

**Figure 4.** Maps of correlations between DARDAR cloud types and SIC for each season (2007-2010).

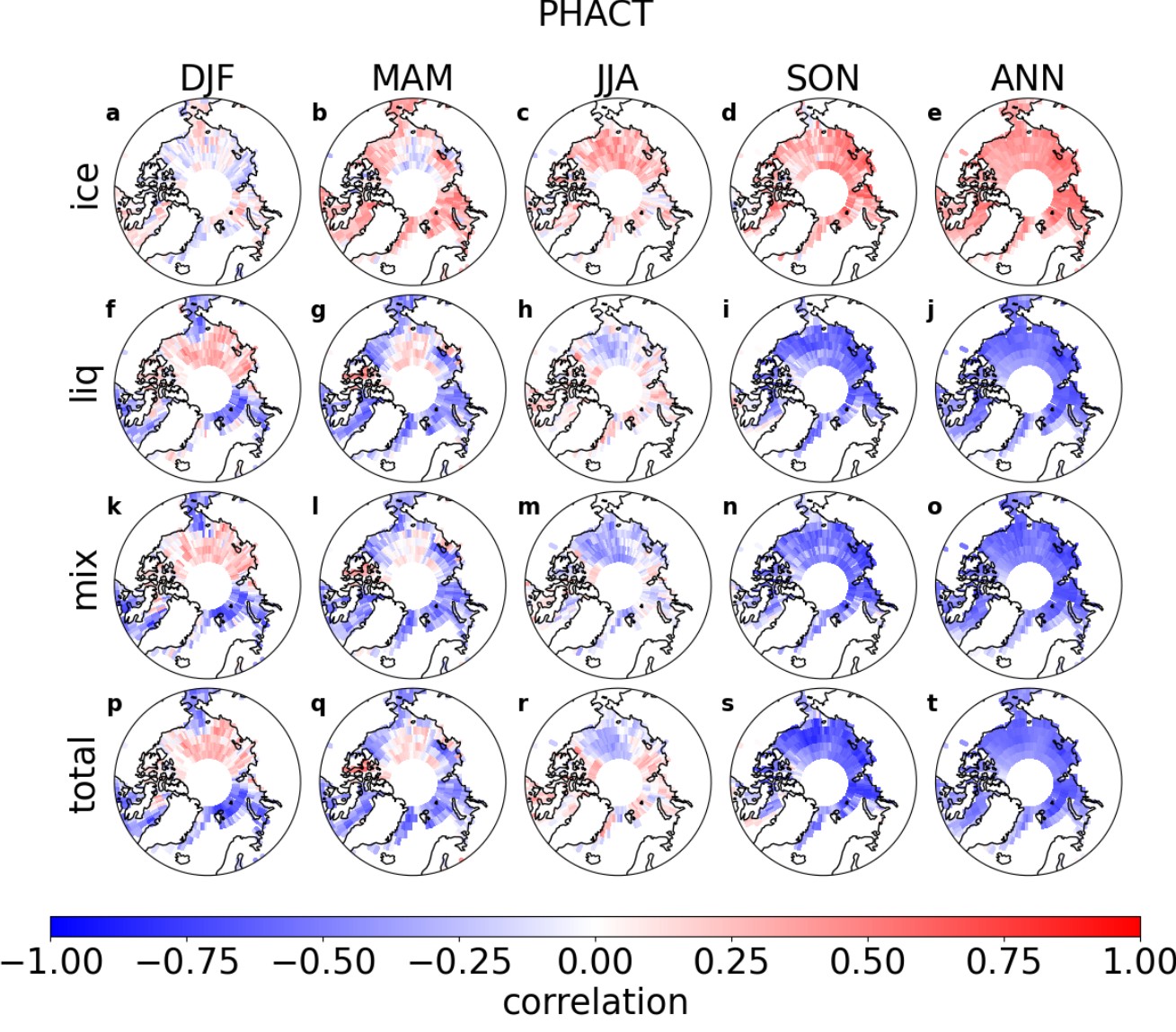

**Figure 5.** Maps of correlations between PHACT cloud types and SIC for each season (2007-2020).

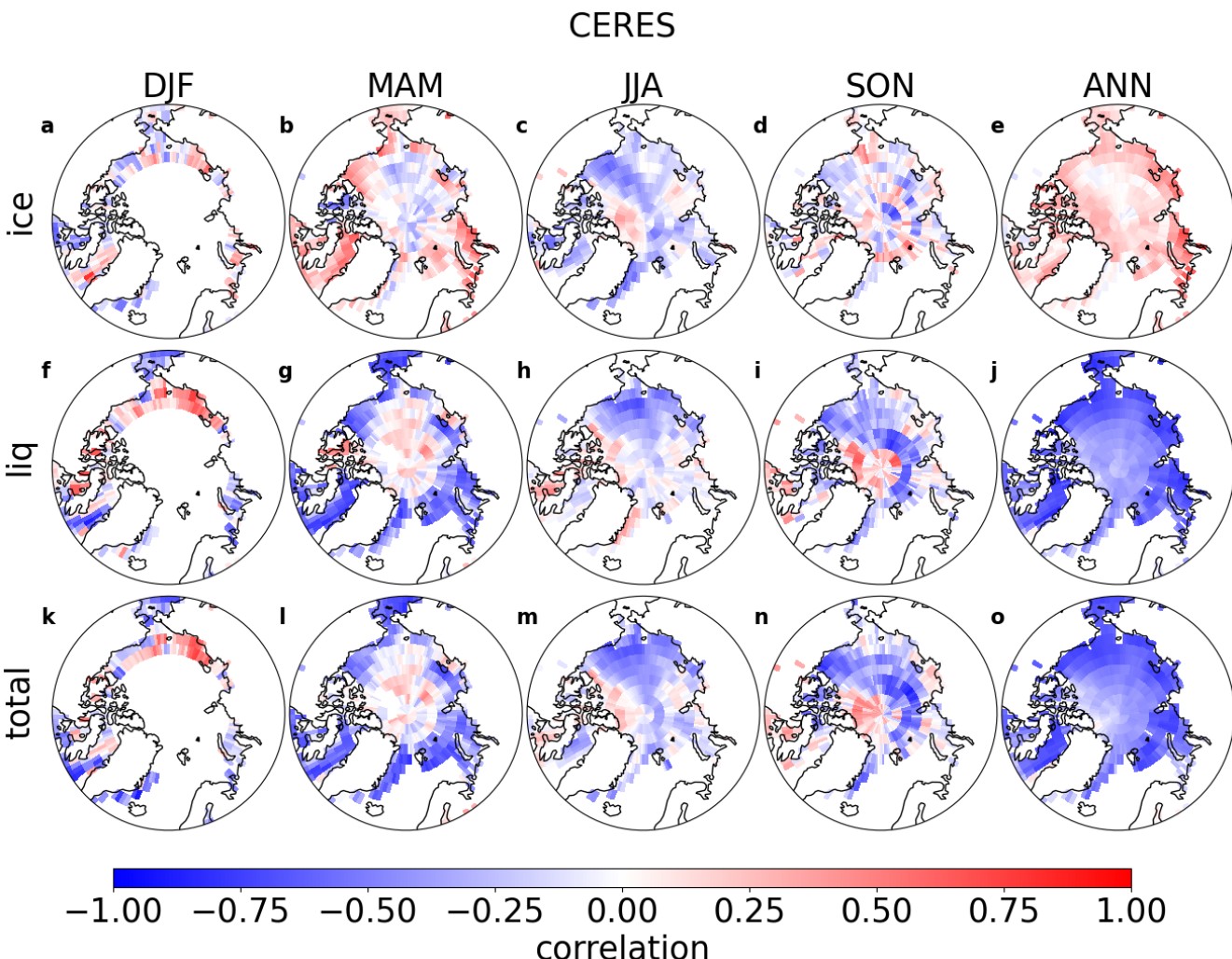

**Figure 6.** Maps of correlations between CERES cloud types and SIC for each season (2007-2020).

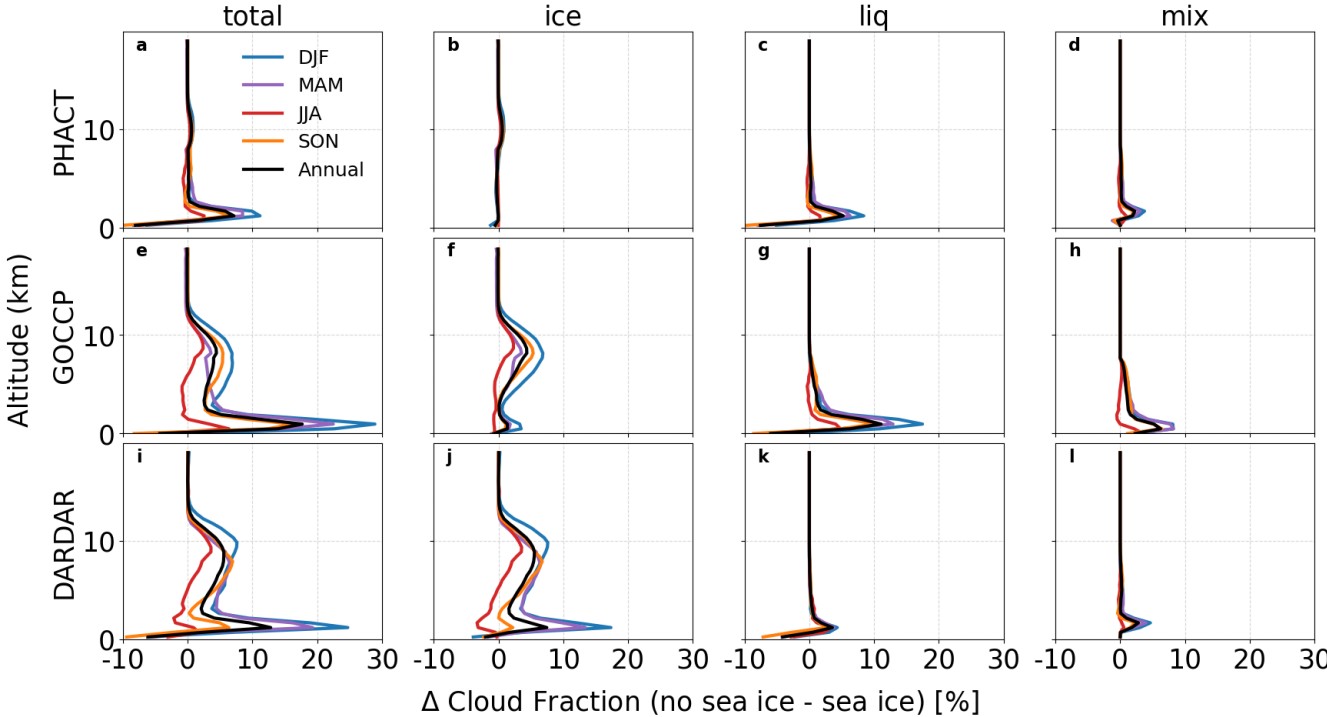

**Figure 7.** Effect of sea-ice conditions on cloud type profiles (%) for PHACT (top, 2007-2020), CALIPSO-GOCCP (middle, 2007-2020) and DARDAR (bottom, 2007-2010)

the season. Despite some disagreements, likely related to the difference in ice cloud definitions, all products agree on positive correlations when using all months. Ice clouds may have two distinct origins: high cirrus clouds mostly controlled by synoptic-scale dynamics through so-called intrusions from lower latitudes (Pithan et al. (2018)), and low and mid-level ice clouds driven by either small convective pockets or local increases in moisture fluxes, which form liquid clouds first and then ice clouds (Pithan et al. (2018)). Over sea ice, both of these processes might be enhanced since the air is generally colder, explaining positive correlations between ice clouds and sea ice.

Next, we take advantage of the active-sensor profiling capability to investigate changes in each cloud-type profile as a function of the surface conditions to determine what altitude contributes the most to cloud variability (Figure 7). Here we compute the difference of cloud type profiles above open-ocean minus sea-ice covered grid boxes, where an open-ocean and sea-ice grid box is defined as SIC < 0.4 and SIC > 0.6, respectively, in order to maximize the number of grid boxes utilized in the computation – using more restrictive thresholds does not change our results (see Fig. S7). Since PHACT only document the uppermost ice cloud level, we also analyze observations from CALIPSO-GOCCP, which documents all cloudy levels and uses the same cloud and phase diagnostics as well as resolution as PHACT. These results are consistent with our correlation analysis and provide additional insights. In the low-levels (below 3 km), the cloud cover change is mostly driven by liquid clouds and the mixed-phase clouds behave like the liquid clouds, which increase over open ocean during all seasons but summer

**Table 2.** Area-weighted changes in longwave, shortwave, and net surface cloud radiative effects ($Wm^{-2}$) between open-ocean and sea-ice grid boxes (CERES, 2007-2020).

| ($Wm^{-2}$) | DJF | MAM | JJA | SON | ANN |
|---|---|---|---|---|---|
| $\Delta LWCRE$ | 7.7 | 10.6 | 2.6 | -0.9 | 5 |
| $\Delta SWCRE$ | -4.4 | -53.3 | -57 | -15.7 | -32.6 |
| $\Delta TotalCRE$ | 3.3 | -42.7 | -54.4 | -16.6 | -27.6 |

(Figure 7c-g-k and d-h-l). In DARDAR and CALIPSO-GOCCP, ice cloud fraction also increases substantially during winter and spring seasons, albeit to a smaller extent in CALIPSO-GOCCP (Figure 7f-j). This is not captured by PHACT (Figure 7b), a product that only document the uppermost ice cloud layer. The boundary layer is the main contributor in PHACT and CALIPSO-GOCCP. In DARDAR however – and in CALIPSO-GOCCP to a smaller extent – the ice cloud fraction increases
substantially over open ocean in mid and high levels in all seasons. Yet the associated ice cloud cover changes are negative in all products, which means that ice clouds are less frequent but more vertically extended over open ocean. Stronger and more frequent convection over open ocean compared to sea-ice surface could very well explain this result.

Finally, in order to determine the radiative impact of these cloud changes, we compute net surface LW, SW and total CRE for open-ocean and sea-ice-covered conditions following the same method as described in the above paragraph (Table 2). Our
results indicate that the increase of cloud cover over open ocean, driven by low-level liquid clouds, corresponds to stronger cooling in all seasons but winter, attributable to larger SW reflection than LW absorption. The correlation maps between SIC and the net effect of the clouds at the surface (Figure 8) are consistent with the correlations found between SIC and liquid cloud covers, albeit of opposite sign. On the one hand, SIC is positively correlated with net SW CRE at the surface (Figure 8f-j) because the reduction in liquid clouds generated by increased SIC allows more SW radiation to be absorbed. On the other
hand, fewer liquid clouds also reduce the net LW CRE warming at the surface, resulting in a negative correlation of net LW CRE and SIC (Figure 8a-e). Yet we note that even when the correlation between clouds and SIC is low (e.g., in summer, Figure 6h), the correlation between net CRE and SIC remains high (Figure 8h), emphasizing the importance of the underlying surface type for surface CRE. Finally, our results show that the correlation of the net total CRE with sea ice is dominated by the SW component for all seasons but winter.

To gain additional insights on surface fluxes, we analyze annual anomalies of SW and LW downwelling, upwelling and CRE fluxes together with SIC over a longer record (2001 – 2020; Figure 9). We find that, as SIC decreases, $SW_{dn}$ decreases because more SW radiation is blocked by clouds (Figure 9a). Simultaneously, $SW_{up}$ decreases because the surface albedo is reduced (i.e., bright sea ice being replaced by dark ocean). Overall, this strengthens the cooling effect of clouds at the surface (i.e., more negative SW CRE), which is consistent with previous literature looking at surface SW CRE trends during spring and summer
(Lelli et al. (2023)). In the LW (Figure 9b), $LW_{dn}$ increases, likely driven by the formation of additional low-level liquid clouds

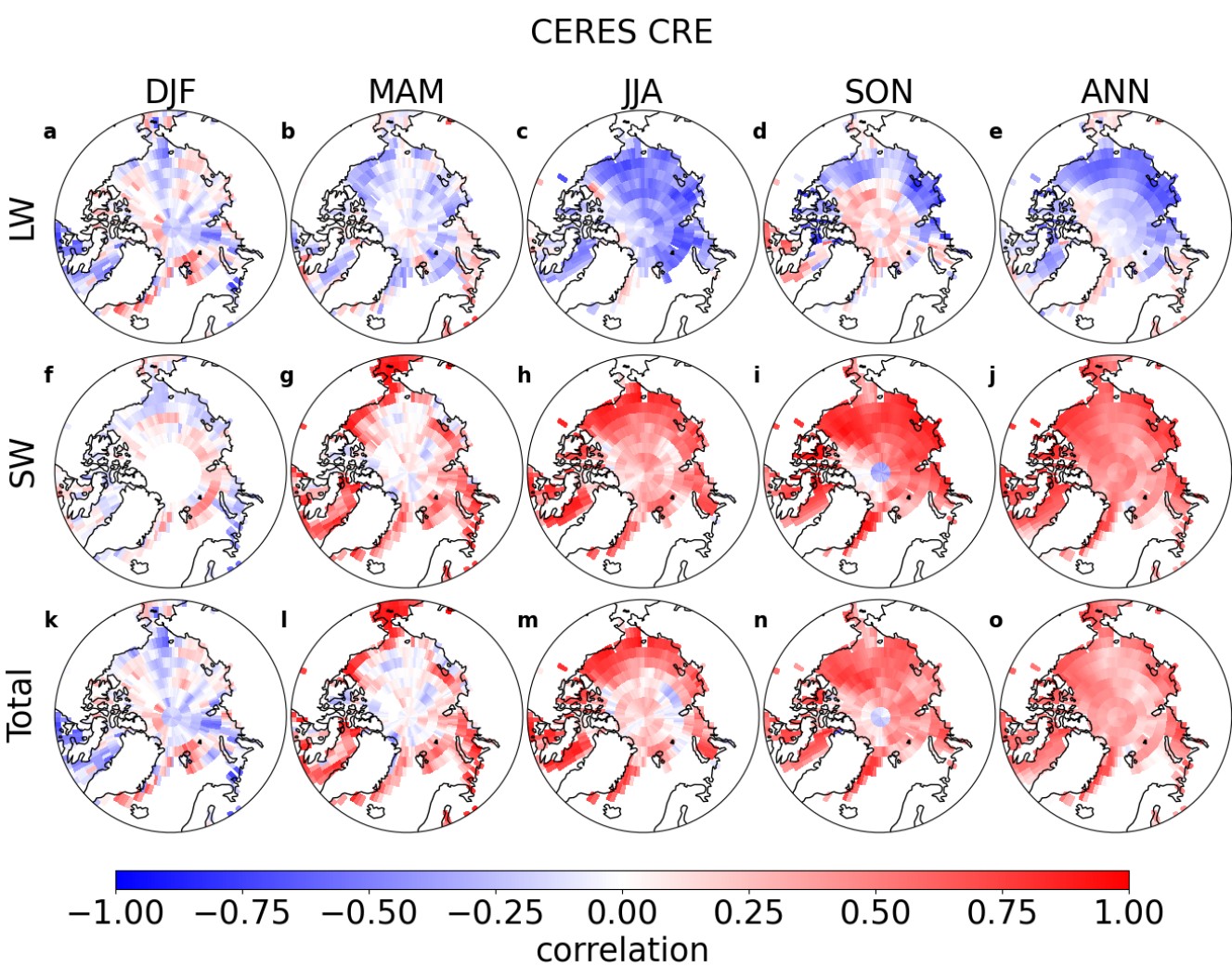

**Figure 8.** Maps of correlations between CERES surface net cloud radiative effects and SIC for each season (2007-2020).

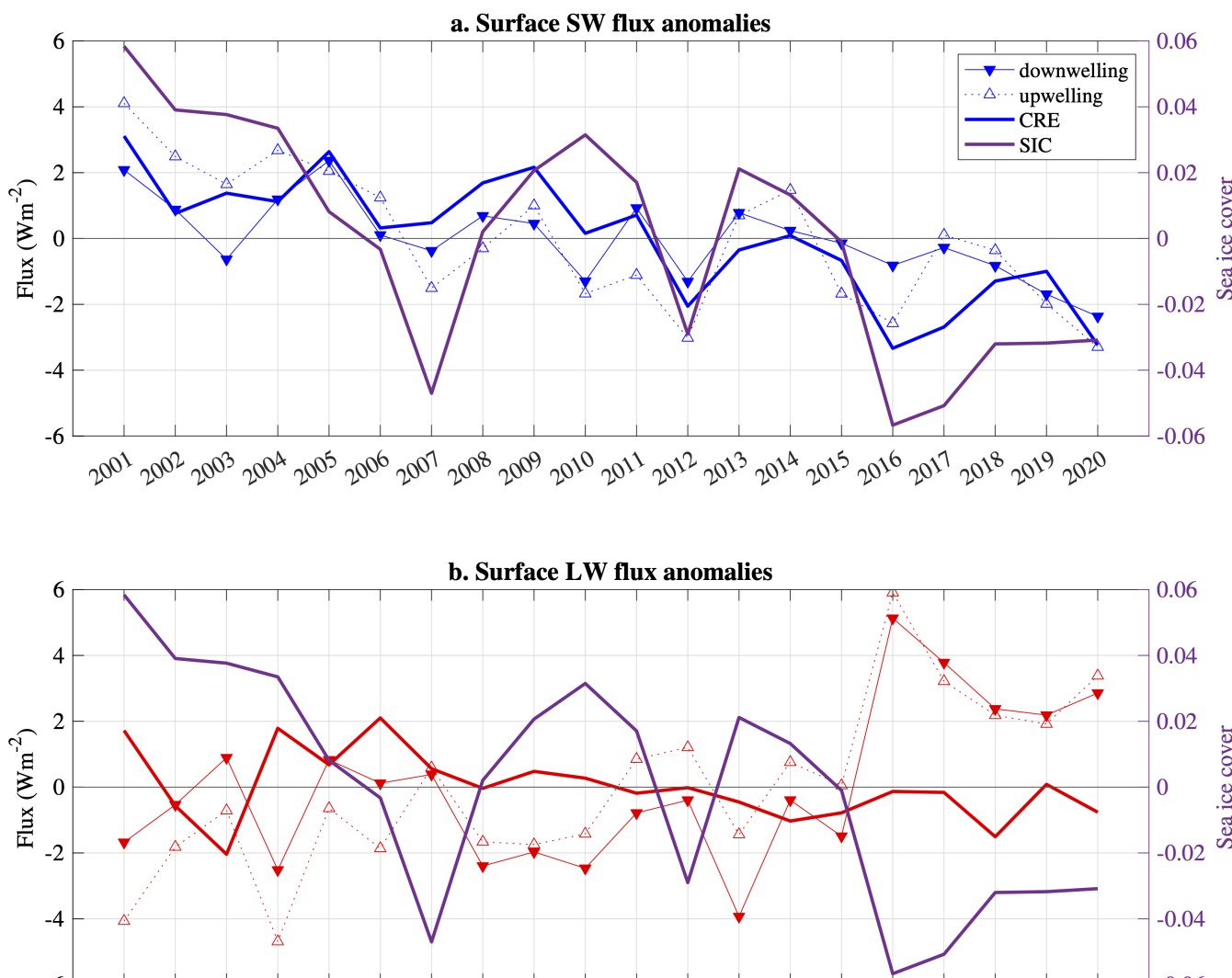

**Figure 9.** Time series of surface SW and LW flux and SIC anomalies from CERES observations and ERA5 reanalysis (2001-2020). The down-welling, up-welling and CRE fluxes correspond to down-pointing triangle, up-pointing triangle and thick solid lines, respectively, for SW (top) and LW (bottom) radiation while SIC is shown in purple.

that emit LW down to the surface, and $LW_{up}$ increases as the surface gets warmer. These changes cancel out each other, which results in a LW CRE that remains neutral.

## 4    Summary and discussion

Using three independent active- and passive-sensor satellite datasets from the A-train, we analyze the relationship between sea ice, cloud phase and surface radiation over the Arctic (north of 60˚N). We find that all three satellite datasets depict a similar pattern and seasonal variability of total and liquid-bearing cloud covers while DARDAR diagnoses far more ice clouds with distinct seasonal variability compared to PHACT and CERES. We then show that the strong negative correlation between Arctic cloud cover and sea ice cover (SIC) is primarily driven by liquid-bearing clouds from the lowest levels (i.e., below 3 km), for which mixed-phase clouds account for 27 % in PHACT and 50 % in DARDAR. This relationship is robust among all satellite observations for all seasons but summer, which is consistent with findings from previous studies using CALIPSO liquid 3D cloud fraction (Morrison et al. (2018)) and CALIPSO-CloudSat total 3D cloud fraction (Taylor and Monroe (2023)). Unlike in other studies though, we find slightly stronger negative correlations between liquid-bearing clouds and SIC during winter than fall, which could be due to winter lower temperatures that make it more difficult for liquid-bearing clouds to form over sea ice. Additionally, our seasonal maps of correlation reveal the presence of regional differences, which may be driven by local processes and synoptic circulation. In particular over the Bering, Barent and Laptev seas, where winter and spring intrusions of low-latitude warm air change the sign of the correlation between liquid cloud cover and SIC. Furthermore, we show that ice-only clouds (DARDAR and PHACT) also correlate well with SIC on average (with some variability depending on the season), which has not been reported in the literature to our knowledge. Finally, the increase of liquid-bearing clouds with open-ocean conditions – and to some extent the decrease of ice clouds – is associated with more radiative cooling from clouds at the surface, attributable to a larger SW CRE cooling than LW CRE warming. Such a cooling effect is found in all seasons but winter, when the LW CRE warming exceeds the SW CRE cooling. A brief analysis of surface flux and SIC anomalies confirm that in response to SIC decrease, liquid-bearing clouds increase, and their associated radiative effect is a cooling of the surface driven by the SW component.

In response to climate warming, Arctic SIC has been declining and will most likely continue to do so in the future, with ice-free summers that could occur as early as 2030 (Kim et al. (2023)). Within this context, it is important to quantify the effect of sea-ice loss on clouds, which are a major contributor to surface radiative budget. Our analysis suggests that optically thick low-level liquid clouds will be more frequent as SIC declines, and this process should contribute to mitigating Arctic surface warming, except in winter. These results could be used to assess the SIC-cloud relationship in climate models, which still struggle to represent cloud phase transition (Cesana et al. (2022)), and thereby help narrow down the large uncertainties in their representation of Arctic amplification (Boeke and Taylor (2018)).

*Data availability.* The PHACT observations will be made available through a Zenodo link when the manuscript is accepted. The DAR-DAR instantaneous files were obtained from https://www.icare.univ-lille.fr/dardar/documentation-dardar-mask/ (restricted access) and the gridded statistics were computed from the ICARE Data and Service Center and a Zenodo link to the gridded files will be made available when the manuscript is accepted. The CALIPSO-GOCCP files were downloaded from (http://climserv.ipsl.polytechnique.fr/cfmip-obs/Calipso_goccp.html. The CERES cloud phase and flux observations were downloaded from the following CERES websites https://ceres-tool.larc.nasa.gov/ord-tool/jsp/FluxByCldTypSelection.jsp and https://ceres-tool.larc.nasa.gov/ord-tool/jsp/SYN1degEd41Selection.jsp, respectively. The ERA5 monthly means of sea-ice cover were downloaded from the Climate Data Store website (https://cds.climate.copernicus.eu/cdsapp#!/d era5-pressure-levels-monthly-means?tab=form).

*Author contributions.* GC designed the study, carried out the analysis, processed PHACT data, and drafted the figures. OP and LV processed the DARDAR and CERES datasets. OP finalized the processing of DARDAR data and prepared the figures and tables. GC and IS developed the PHACT dataset. GC wrote the manuscript with contributions from all co-authors.

*Competing interests.* The authors declare that they have no conflict of interest.

*Acknowledgements.* GC, MO and ZJ were supported by National Aeronautics and Space Administration IDS (Interdisciplinary Research in Earth Science) (80NSSC20K1523) (award 19-IDS19-0059). GC was also supported by a CloudSat-CALIPSO RTOP at the NASA Goddard Institute for Space Studies. GC and IS were also supported by the U.S. Department of Energy's (DOE) Atmospheric System Research, an Office of Science Biological and Environmental Research program, under grant DE-SC0021004. We would also like to thank the NASA Office of STEM Engagement at GISS, the Minority University Research and Education Project (MUREP), and the NASA OSTEM Internship program, for the support offered to OP and LV. Resources supporting this work were provided by the NASA Center for Climate Simulation (NCCS) at Goddard Space Flight Center. We thank NASA and CNES for giving access to CALIPSO observations, and Climserv for giving access to CALIPSO-GOCCP observations and for providing computing resources.

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
