# Peer review of "The correlation between Arctic sea ice, cloud phase and radiation using A-train satellites"

_EGUsphere, 2023_

## Referee Comment (RC1)

**Overview**
By utilizing three satellite data products, the authors are trying to explore the relationship between the sea ice, clouds, and radiation in the Arctic. Though the results are within expected, studies like this work are important to gain a better/deeper understanding of the highly coupled Arctic system, especially from the observational perspective. I think this paper can be published on ACP after addressing the comments listed below.

**Major comments**

1. About the relationship table (Fig 3.), I have a couple of questions. (a)In CERES data, we see a negative SIC- ice cloud relation is all seasons, yet a positive relationship in terms of the average. I notice the authors mentioned this a bit in the text, I wonder how reliable are these numbers? (b) Mixed-phase clouds have a stronger negative relation with SIC than liquid clouds, especially when looking at non-summer seasonal correlation. Can the authors explain why? (c) related to (c), for DARDAR, the average correlation for liquid clouds is greater than mixed-phase clouds (-0.81 versus -0.67), while seasonal correlation is the opposite. Why? Any possibility this is due to the limited data [4 years only]? If so, the authors should be more cautious when interpreting the correlation numbers here and explain why.
2. By looking at the maps in Fig5, the strongest correlation happens in Autumn (SON), not winter (DJF), which is opposite to these numbers indicated in Fig3. This is especially true for the PHACT data. Please reconcile.
3. Why is the ice cloud much more over the ocean than over sea ice in Fig.7? Is this due to the deeper convection and higher possibility of the cirrus clouds or low- and mid-level ice clouds driven by small local convection? On the other hand, we see little difference (open water - sea ice) in terms of the liquid and mixed-phase clouds, both of which are low-lying clouds and seem independent from the surface type (open ocean vs sea ice). These seem not consistent with the author's interpretation in the text (also see minor #3).
4. In addition to the maps in Fig 8, I am thinking it would be very helpful to show a time series of LW_up, LW_dn, SW_up, SW_dn, and LW_CRE and SW_CRE at the surface, such as during 2007 - 2020 when sea ice is observed to decline. This would not only provide detailed process-level understanding, rather than the mean, but also offer insights for the future Arctic change.

**Minor comments**

1. L10: In the abstract, it would be better to briefly mention why SIC decreases + liquid cloud increase result in an enhanced surface cooling at the surface.

2. L145: Are the results sensitive to the SIC threshold to classify open-water (SIC < 0.4) and sea-ice (SIC > 0.6) grid boxes?
3. L150: For DARDAR, ice cloud cover increase over sea ice is not subtle; instead, it is significant (>10% for DJF below 5 km) and largely dominate the total cloud cover change.

4. Fig.2: liquid clouds have the largest seasonality; I wonder if the community knows the reason for this. Also, the pure liquid cloud cover is larger than mixed-phase cloud cover; is this consistent with other observational studies? Can you explain why this is the case?

---

## Author Comment (AC1)

Dear Editor,

We thank you, the two reviewers and Luca Lelli for their time and effort in carefully reading our manuscript and providing useful feedback. Our point-by-point response to each reviewer and community comment is provided below in blue. The line numbers refer to the highlighted version of the manuscript. In summary, the main changes are:

1) We have explained the inconsistencies in correlation computations between maps and tables and among datasets, and we have investigated time series of surface fluxes and SIC anomalies (Reviewer 1)

2) We have clarified the method and added sensitivity analyses on the thresholds we use throughout the study (Reviewer 2),

3) We have expanded our description of the datasets and associated uncertainties, and we have added references to previous work regarding surface flux trends (Luca Lelli).

Additionally, we have added Israel Silber as a coauthor of this study because of his contribution to the development of the PHACT dataset.

**Reviewer 1**

By utilizing three satellite data products, the authors are trying to explore the relationship between the sea ice, clouds, and radiation in the Arctic. Though the results are within expected, studies like this work are important to gain a better/deeper understanding of the highly coupled Arctic system, especially from the observational perspective. I think this paper can be published on ACP after addressing the comments listed below.

**Major comments**

1. About the relationship table (Fig 3.), I have a couple of questions. (a)In CERES data, we see a negative SIC- ice cloud relation is all seasons, yet a positive relationship in terms of the average. I notice the authors mentioned this a bit in the text, I wonder how reliable are these numbers? (b) Mixed-phase clouds have a stronger negative relation with SIC than liquid clouds, especially when looking at non-summer seasonal correlation. Can the authors explain why? (c) related to (c), for DARDAR, the average correlation for liquid clouds is greater than mixedphase clouds (-0.81 versus -0.67), while seasonal correlation is the opposite. Why? Any possibility this is due to the limited data [4 years only]? If so, the authors should be more cautious when interpreting the correlation numbers here and explain why.

a) The main reason to explain this discrepancy is that the average correlation is a temporal correlation between Arctic SIC and cloud cover time series as opposed to more general spatial/temporal correlations for the seasons. This means that the synoptic environmental conditions associated with the presence of sea ice across the Arctic favor ice clouds and hinder liquid-bearing cloud formation. However, on a finer scale perspective, parameters other than sea ice can influence the formation of clouds (e.g., Taylor and Monroe 2023, Morrison et al., 2018), which explains why the general correlations using all grid boxes and time increments are smaller. Regarding the specific CERES dataset, the spatial correlations for ice clouds are weaker than the other two datasets because it is impossible to completely isolate the ice clouds from liquid-bearing clouds since MODIS only detects cloud tops. As a result, some liquid clouds may occur concurrently with MODIS ice-diagnosed clouds and affects sea-ice cloud correlations. We have added some explanation in the manuscript (L162-169): "*More surprisingly, our analysis indicates that the ice cloud cover is somewhat positively correlated with SIC in all datasets, though less so in CERES likely because of liquid cloud contamination, which, to our knowledge, has not been reported in the literature before. It is also interesting to note that these correlations are stronger when correlating Arctic-averaged time series of SIC and cloud covers (Fig. 2 last column). This discrepancy might be indicative of the influence of local parameters other than sea ice on clouds; when looking at a pan-Arctic perspective, this local influence fades out and the presence of sea ice is associated with synoptical meteorological conditions that favor ice cloud formation and hinders that of liquid clouds.*"

b) This is indeed an interesting point. By design, mixed-phase clouds are more opaque clouds in both GOCCP and DARDAR. These opaque clouds are more mature liquid-bearing than the more optically thin liquid clouds, which are the early stage of liquid cloud lifecycle. Concurrently, we find that liquid clouds are favored by open-ocean conditions while sea-ice covered surface tend to prevent liquid cloud formation. To get to that mature state, the opaque liquid-bearing clouds must have been lingering over open-ocean for a longer time than the thin ones, likely explaining why the correlation between mixed-phase clouds and open ocean is stronger. Lidar-based literature supports the fact that optically thin and relatively short-lived liquid clouds (compared to opaque clouds) tend to be prevalent over sea-ice covered surfaces (Lacour et al., 2018; Silber et al., 2020). We have added some explanation in the manuscript (L182-186): "*Furthermore, we find that the correlations are even stronger for mixed-phase clouds than for liquid clouds in PHACT. In this dataset, the mixed-phase cloud are more opaque than liquid clouds. Recent literature suggests that opaque clouds are at a more mature stage of their lifecycle than thin clouds (Silber et al. 2020), and therefore, they would have to linger over open ocean for longer to reach that mature stage, which could explain the*

*stronger correlations of mixed-phase clouds with open ocean compared to optically thinner liquid clouds.*"

c) Using 4 years with PHACT provides the same results. It could be related to the diagnostic of mixed-phase clouds in DARDAR.

2. By looking at the maps in Fig5, the strongest correlation happens in Autumn (SON), not winter (DJF), which is opposite to these numbers indicated in Fig3. This is especially true for the PHACT data. Please reconcile.
What happens here is that the strength of the total correlation reported in Fig. 3 (the colored table) is dictated by grid boxes in which SIC varies a lot whereas the grid boxes in which SIC is close to 1 or 0 do not affect the correlation as much. However, even though the negative correlation is high in DJF, in some grid boxes, it is positive, and these grid boxes happen to be where SIC is always equal to 1, vary poorly and do not affect the total correlation as much (see Fig. R1 below). When using more restrictive SIC thresholds, it makes the positive correlations go away and DJF total correlation is still larger in magnitude than SON's, as shown in Fig. R2 below. Then why are DJF correlations larger than SON's? We suspect that it is more difficult for liquid clouds to form over sea ice in winter when temperatures are colder and therefore liquid clouds have a stronger relationship with sea ice during that season. We have mentioned that in the manuscript (L177-180): "*Since these correlations occur mostly over grid boxes in which SIC is close to 1 – and does not vary much – the total correlation (Figure 3) remains highly negative, even more so than in SON. We speculate that it is more difficult for liquid clouds to form over sea ice than it would be in fall because of the lower temperatures – for similar environmental conditions – hence the stronger negative correlations in winter.*"

And in the conclusion (L244-246): "*Unlike in other studies though, we find slightly stronger negative correlations between liquid-bearing clouds and SIC during winter than fall, which could be due to winter lower temperatures that make it more difficult for liquid-bearing clouds to form over sea ice.*"

[Figure]

*Figure R1: Scatter plots (left) of SIC and liquid cloud cover for mean SIC thresholds between 0.01 and 0.99 (same as in the manuscript) and maps of correlations (right) for DJF (top) and SON (bottom). The red dots correspond to grid boxes in which the correlation is greater than 0.4.*

[Figure]

*Figure 2: Same as R1 for mean SIC thresholds between 0.1 and 0.9.*

3. Why is the ice cloud much more over the ocean than over sea ice in Fig.7? Is this due to the deeper convection and higher possibility of the cirrus clouds or low and mid-level ice clouds driven by small local convection? On the other hand, we see little difference (open water - sea ice) in terms of the liquid and mixed-phase clouds, both of which are low-lying clouds and seem independent from the surface type (open ocean vs sea ice). These seem not consistent with the author's interpretation in the text (also see minor #3).

The magnitude of the change is greater in the low levels in all datasets. However, the change in ice clouds is substantial in DARDAR. We have added CALIPSO-GOCCP in this figure because it documents clouds at all levels, rather than just the uppermost level in PHACT, and it's consistent with PHACT, since PHACT uses CALIPSO-GOCCP instantaneous files. We find that the change in ice clouds is also non negligible but rather small compared to DARDAR. Yet it shows that ice clouds are more vertically extended but less spread out over open ocean than over sea ice, which could be because of convection indeed. We've added some discussion about these (L204-214): "*In DARDAR and CALIPSO-GOCCP, ice cloud fraction also increases substantially during winter and spring seasons, albeit to a smaller extent in CALIPSO-GOCCP. This is not captured by PHACT, a product that only document the uppermost ice cloud layer. The boundary layer is the main contributor in PHACT and CALIPSO-GOCCP. In DARDAR however – and in CALIPSO-GOCCP to a smaller extent – the ice cloud fraction increases substantially over open ocean in mid and high levels in all seasons. Yet the associated ice cloud cover changes are negative in all products, which means that ice clouds are less frequent but more vertically extended over open ocean. Stronger and more frequent convection over open ocean compared to sea-ice surface could very well explain this result.*"

4. In addition to the maps in Fig 8, I am thinking it would be very helpful to show a time series of LW_up, LW_dn, SW_up, SW_dn, and LW_CRE and SW_CRE at the surface, such as during 2007 - 2020 when sea ice is observed to decline. This would not only provide detailed process-level understanding, rather than the mean, but also offer insights for the future Arctic change.

To address the reviewer's comment, we have computed annual anomalies of all these quantities from 2001 to 2020. We found that SWdn is decreasing because more SW is blocked by clouds and at the same time, SWup is decreasing because SIC is decreasing and thereby reflects less SW back up. In the LW, as SIC decreases, LWdn increases likely because there are more clouds that emit LW down to the surface, and LWup increases likely because the surface gets warmer. This results in a LWCRE that remains neutral.

*We have added the analysis in the manuscript (L227-234): "To gain additional insights on surface fluxes, we analyze annual anomalies of SW and LW downwelling, upwelling and CRE fluxes together with SIC over a longer record (2001 – 2020; Fig. 9). We find that, as SIC decreases, $SW_{dn}$ decreases because more SW radiation is blocked by clouds. Simultaneously, $SW_{up}$ decreases because the surface albedo is reduced (i.e., bright sea ice being replaced by dark ocean). Overall, this strengthens the cooling effect of clouds at the surface (i.e., more negative $CRE_{SW}$). In the LW, $LW_{dn}$ increases, likely driven by the formation of additional low-level liquid clouds that emit LW down to the surface, and $LW_{up}$ increases as the surface gets warmer. These changes cancel out each other, which results in a LW CRE that remains neutral."*

*And in the conclusion (L253-255): "A brief analysis of surface flux and SIC anomalies confirm that in response to SIC decrease, liquid-bearing clouds increase, and their associated radiative effect is a cooling of the surface driven by the SW component."*

**Minor comments**
1. L10: In the abstract, it would be better to briefly mention why SIC decreases + liquid cloud increase result in an enhanced surface cooling at the surface.
*We have modified the last two sentences of the abstract to include this information (L11-15): "Our findings indicate that the newly formed liquid clouds reflect more shortwave (SW) radiation back to space compared to the surface, generating a cooling effect of the surface, while their downward longwave (LW) radiation is similar to the upward LW surface emission, which has a negligible radiative impact on the surface. This overall cooling effect should contribute to dampening future Arctic surface warming as SIC continues to decline."*

2. L145: Are the results sensitive to the SIC threshold to classify open-water (SIC < 0.4) and sea-ice (SIC > 0.6) grid boxes?
*Using more restrictive thresholds such as 0.15 and 0.85 doesn't change the results qualitatively, although there is small difference in the magnitude as shown in new Fig. S7. We have added the information in the manuscript (L199): " – using more restrictive thresholds does not change our results (see Fig. S7)."*

3. L150: For DARDAR, ice cloud cover increase over sea ice is not subtle; instead, it is significant (>10% for DJF below 5 km) and largely dominate the total cloud cover change.
*We have addressed this comment in main comment 3.*

4. Fig.2: liquid clouds have the largest seasonality; I wonder if the community knows the reason for this. Also, the pure liquid cloud cover is larger than mixedphase cloud cover; is this consistent with other observational studies? Can you explain why this is the case?
*The liquid cloud seasonality has been documented before (Cesana et al., 2012; Lacour et al., 2017; Mioche et al., 2015; McIlhattan et al., 2017; Shupe, 2011). It is mostly due to environmental conditions that favor liquid cloud formation, i.e., more moisture and mild temperature in summer and transition seasons as opposed to winter. We have added this sentence to clarify (L144-146): "That seasonality of liquid clouds is mainly attributable to environmental conditions that favor liquid cloud formation, i.e., more moisture and mild temperature in summer and transition seasons as opposed to winter."*

**Reviewer 2**
This study compares three different datasets of cloud fraction from satellites, including partitioning by phase in the Arctic. The responses of cloud fraction and cloud radiative effect to changes in sea ice concentration (SIC) are quantified. Much of the results are consistent with past work, e.g., the seasonality of correlations between SIC and liquid-containing clouds. Unique to this work is comparing results from active and passive sensors, as well as quantifying the response of ice clouds to SIC in the analysis.

**Main comments**:
Other works have used quite specific methods to isolate the response of clouds to SIC changes and avoid confusing the cloud response to sea ice with the cloud response to atmospheric variability, e.g., the intermittent mask in Morrison et al. (2018) and the marginal ice zone crossing events in Taylor and Monroe (2023), along with instantaneous observations. First, this work uses some thresholding for grid cells based on SIC in Figure 7 and associated conclusions. Are the results, particularly for ice clouds, sensitive to this choice? Second, after (re)reading the Datasets section and data availability statement, it's not actually clear to me what time resolutions the data used are. I assumed with using CALIPSO and CloudSat products the observations would be instantaneous, but the ERA5 SIC are monthly means. Could you please clarify this in the Datasets section?

In our study, we look at the overall behavior of clouds and sea ice and aim at providing some constraint for models on climate scale rather than trying to better understand the fine-scale processes at play between sea ice and clouds. These studies have shown that overall, clouds are sensitive to meteorological conditions associated to the presence of sea ice, rather than sea ice itself, which prevent liquid cloud formation. We have slightly modified the introduction to clarify (L32-33): *"In general, these studies try to better understand the relationship between sea ice and clouds at the process level and show that clouds are most likely controlled by air-surface coupling. However, they focus on...*", and (L36-37) *"In this study, we build on their results to analyze the relationship between sea ice and clouds on longer time scales to advance our understanding of Arctic climate and with the hope to provide constraint for climate models*."

Regarding the method,
1) We pick a threshold of 0.01 and .99 to avoid fully open-ocean and sea-ice-covered conditions. We can be slightly more restrictive by using 0.02 and 0.98 and 0.03 and 0.97 and this doesn't affect qualitatively our results, as shown in the table below. This table shows the annual mean temporal and spatial correlations for the different thresholds. Using 0.01 and 0.99 also allows to maximize the number of points used to compute the correlations, as shown in the Figure below. We are now mentioning this in the manuscript (L157-159): "*that is averaged SIC is greater than 0.01 (excluding open-ocean only grid boxes) and smaller than 0.99 (excluding sea-ice-covered only grid boxes). Using a more restrictive thresholds (0.02 and 0.98 or 0.03 and 0.97) does not affect our results qualitatively (Table S1)."*

2) Although CALIPSO and DARDAR files used are instantaneous files, they are then averaged onto 2.5x2.5˚ and 1x1˚ grids. Then all datasets are interpolated to 2.5x2.5˚ grids, as specified line 93 of the original manuscript. We have added a line break to make this information more visible. We have also added the information about PHACT, DARDAR and CERES grids.

I was taught to be suspicious of cloud products from passive sensors in the Arctic, especially with changes in SIC, due to better detectability over open ocean compared to clouds (e.g., Liu et al. 2010). Is that not a concern with the CERES dataset used here, or could it impact any of the results?
This was a concern going in this study using CERES cloud product. However, we can see that CERES (MODIS) observations are doing a good job at capturing the cloud phase and the relationship of clouds with SIC although there are some inconsistencies compared to the active-sensor products. However, we also note that cloud phase is available only during daytime, which is a limiting factor, especially when

looking at winter statistics. We already mention this limiting factor in the data section, but we have added a sentence about CERES doing a surprisingly great job at capturing this relationship in the result section (L121-123): *"It is worth noting that although CERES observations come from a passive sensor instrument, which makes it more challenging to detect clouds and retrieve their properties over bright surfaces, they are in good agreement with the other two active-sensor observations."*

Minor comments:
To some extent this is personal preference, but adding subplot labels and referencing them in the text might help make it easier for readers to connect the text to specific figures.
Done.

Line 18: More recently Middlemas et al (2020) also found that cloud feedbacks have little impact on Arctic amplification.
We have added this reference.

Line 60: 20% seems like a substantial difference, no?
This a maximum fraction difference for the phase, but it is actually 11.8 and 1% for two collocated in situ flights in the Arctic, when accounting for uncertainty in the in situ discrimination thresholds. We have modified the sentence (L70-72): "*A validation study against in situ aircraft measurements shows that the maximum disagreement fraction between CALIPSO-GOCCP instantaneous profiles – used in PHACT – and five in situ aircraft flights is 11.8% when accounting for in situ aircraft measurement uncertainty (Cesana et al. 2016; their Table 3)."*

Line 85-6: Does "large agreement fraction with...phase retrievals" mean the two datasets agree well on the phase or also the amount of clouds?
This refers to cloud phase for pixels that are diagnosed as cloudy in both datasets. We have modified the sentence to clarify: We note that MODIS-C6 cloud phase has a large agreement fraction with CALIPSO science team cloud phase retrievals (up to 90% Marchant et al. 2016) – for clouds that are detected by both instruments (L101): "*We note that MODIS-C6 cloud phase has a large agreement fraction with CALIPSO science team cloud phase retrievals (up to 90%, Marchant et al. 2016) – for clouds that are detected by both instruments…*"

Line 96: I don't see a sharp contrast between land and ocean in DARDAR on the Pacific side of the Arctic. To me it looks like North America, Russia, and the ocean are all in the 70-80 CF bin?
We meant the for the liquid bearing clouds. We have corrected this in the manuscript (L116): "*Interestingly, the datasets also detect a very similar liquid cloud cover (in terms of means and pattern correlations, Table 1 and Fig. S3), with a sharp contrast between land and sea and maxima over the Laptev, Barent and Greenland seas*".

Lines 106-8: Why the large difference between ground-based and space-based measurements? Is it a difference in location, land for ground-based but a mix of land and ocean for space-based?
These discrepancies are attributable to lidar attenuation in the case of PHACT, which does not allow the lidar beam to reach the mixed-phase layer, to ground clutter in the case of CloudSat-based measurements, which prevents retrieving hydrometeors below ~500 m, and to spatial variability (i.e., one data point in ground-based measurements against an Arctic-wide average in spaceborne measurements). We now mention this in the revised manuscript (L129-132): "*These discrepancies are attributable to lidar attenuation in the case of PHACT, which does not allow the lidar beam to reach the mixed-phase layer, to ground clutter in the case of CloudSat-based measurements, which prevents retrieving hydrometeors below ~500 m, and to spatial variability (i.e., one data point in ground-based measurements against an Arctic-wide average in spaceborne measurements)."*

Line 109: "show more"
Done

Lines 112-3: Why does the CERES product have more ice cloud fraction than PHACT?
PHACT is showing ice only detection, meaning that if liquid is detected below, the ice cloud detection is disregarded for this diagnostic, while CERES reports the topmost ice cloud fraction without information of layers underneath. When accounting for all ice clouds that are above liquid clouds, as CERES would see them, then PHACT ice cloud cover is larger than CERES's as shown in the figure below. We now explain this in the manuscript (L138-140): "*CERES reports larger ice cloud cover than PHACT because it does not diagnose ice-only cloud column as in PHACT. When accounting for ice above liquid clouds as well as ice-only clouds, consistent with CERES observations, PHACT detects more ice clouds than CERES (Fig. S5).*"

[Figure]

Lines 118-9: There are still differences in magnitude, and some slight differences in seasonality (e.g., some minima occur in March and April). Are you able to explain these differences?
The difference is mostly between CERES and the other and is most likely related to the use of daytime only in CERES, which restricts the latitudes during winter months. We have modified the manuscript to clarify (L146-148): "*Yet we note some small differences between CERES and the CALIPSO-based observations during the winter months, most likely because of the use of daytime only observations in CERES, which restricts the latitudes during these months.*"

Figure 2: I'm not sure if it would show a meaningful difference, but do the seasonal cycles from CERES and PHACT change is you compare the same years as DARDAR?
We have reproduced the figure for the two different periods for CERES and PHACT. The results are nearly identical as shown below, which is now specified in the manuscript (L149): "*Using 2007-2010 instead of 2007-2020 does not change qualitatively or quantitatively our results (Fig. S6).*"

[Figure]

Lines 128-9: Shouldn't the cloud response to SIC be regional/local? Is something else going on if the response is larger for Arctic-wide averages?

*The synoptic meteorological conditions associated with the presence of sea ice over the Arctic favor ice clouds and hinder liquid-bearing cloud formation. However, on a finer scale perspective, parameters other than sea ice can influence the formation of clouds (e.g., Taylor and Monroe 2023, Morrison et al., 2018), which explains why the map correlations (Fig. 4-6) computed using all grid boxes are smaller. We have expended our explanation in the manuscript (L166-169):* "*This discrepancy might be indicative of the influence of local parameters other than sea ice on clouds; when looking at a pan-Arctic perspective, this local influence fades out and the presence of sea ice is associated with synoptical meteorological conditions that favor ice cloud formation and hinders that of liquid clouds.*"

Line 140: In DARDAR winter and summer the maps don't look obviously positive or negative to me but more like noise.

*We have modified this sentence (L188) such as:* "*DARDAR correlations are consistent with those form PHACT in spring and fall while CERES correlations are both negative and positive depending on the season.*"

Line 141: Do you have an explanation for the different responses of ice clouds to SIC in the different datasets?

*This is most likely related to differences in ice cloud definitions. For example, DARDAR has a tendency to overestimates ice clouds and some of the CERES ice clouds may contain liquid clouds underneath as opposed to PHACT and DARDAR. We have added (L189-190):* "*DARDAR correlations are consistent with those form PHACT in spring and fall while CERES correlations are both negative and positive depending on the season. Despite some disagreements, likely related to the difference in ice cloud definitions, all products agree on positive correlations when using all months.*".

Line 149: Are the results sensitive to the SIC thresholds used, specifically if you used more strict definitions of open ocean (e.g., SIC<15%) and ice-covered (e.g., SIC>80%)? Also, why these values? I might expect SIC near 0.4 and 0.6 to have mixed influences of ocean and sea ice.

*There are no qualitative differences when choosing more conservative thresholds, however the magnitude of the change is slightly different. We picked these thresholds to maximize the sample number. We have added this information in the manuscript (L199):* "*– using more restrictive thresholds does not change our results (see Fig. S7)*"

Lines 153-4: I see that total CF changes look like liquid CF changes in PHACT (upper row), but to me it appears the DARDAR has more changes in ice (bottom row, second from the left)? Am I mis-interpreting the figure? Is there an explanation for largest changes in ice CF in DARDAR?

*This is correct, and this is attributable to two main reasons: i) DARDAR may overestimate ice cloud fraction since it uses CALIPSO level2 product to detect thin cirrus clouds and it diagnoses ice clouds when the lidar signal is fully attenuated and the temperature is below 0˚C, and ii) all clouds are documented in the profile whereas in PHACT, only the uppermost ice clouds is documented. We find a larger change in ice clouds when we use CALIPSO-GOCCP ice cloud profiles – a dataset that also documents all clouds rather than the uppermost ones – in better agreement with DARDAR. We have modified the text to explain this (L199-201):* "*Since PHACT only document the uppermost ice cloud level, we also analyze observations from CALIPSO-GOCCP, which documents all cloudy levels and uses the same cloud and phase diagnostics as well as resolution as PHACT.*" *And later on (L204-214):* "*In DARDAR and CALIPSO-GOCCP, ice cloud fraction also increases substantially during winter and spring seasons, albeit to a smaller extent in CALIPSO-GOCCP. This is not captured by PHACT, a product that only document the uppermost ice cloud layer. The boundary layer is the main contributor in PHACT and CALIPSO-GOCCP. In DARDAR however – and in CALIPSO-GOCCP to a smaller extent – the ice cloud fraction increases substantially over open ocean in mid and high levels in all seasons. Yet the*

*associated ice cloud cover changes are negative in all products, which means that ice clouds are less frequent but more vertically extended over open ocean. Stronger and more frequent convection over open ocean compared to sea-ice surface could very well explain this result.*"

Lines 160-164: How much of the SW difference is changing clouds vs different surface albedo? Like you say, SW CRE can change just based on the surface albedo changing even if clouds remain the same.
It is correct that the CRE would not be the same over different surface types for similar cloud properties. However, in our case, the CERES CRE is calculated assuming the same surface type on the all-sky and clear fluxes, so it doesn't affect our results. We have added this information in the description of the datasets (L94-96): "*The computation of all-sky and clear-sky fluxes assumes the same surface type, which means that the surface albedo is accounted for in the computations of CRE values (Section 3)*".

Line 171: Did lines 111-3 mean that the larger ice CF in DARDAR is inaccurate?
Correct, it can be inaccurate in some instances because of the along-track averaging. For example, the CALIPSO science team product, which is based on the same level2 product that is used in DARDAR, overestimates the cirrus cloud fraction by up to 24% compared to in situ aircraft measurements at mid-latitudes (Cesana et al., 2016).

Line 180: The correlation between ice clouds and SIC seems dependent on season and dataset – maybe add some qualifiers to this statement?
We have modified the sentence to address the reviewer's comment (L249-250): "*we show that ice-only clouds (DARDAR and PHACT) also correlate well with SIC on average (with some variability depending on the season),*"

Line 184: Extra "and" between "declines and will"
Here we meant that the SIC declines in the present-day climate "and" will continue to decline in the future. We have modified the sentence to make it clearer (L256): "*In response to climate warming, Arctic SIC has been declining and will*"

References:
Liu, Yinghui, et al. "Errors in cloud detection over the Arctic using a satellite imager and implications for observing feedback mechanisms." Journal of Climate 23.7 (2010): 1894-1907.
Middlemas, E. A., et al. "Quantifying the influence of cloud radiative feedbacks on Arctic surface warming using cloud locking in an Earth system model." Geophysical Research Letters 47.15 (2020): e2020GL089207.

**Community comment by Luca Lelli**

This commentary is not intended to be a comprehensive review but rather to comment on some aspects that, as a cloud researcher in the Arctic myself, I find interesting or worthy of improvement and explanation. My main points are three: the PHACT dataset, the error treatment and the conclusions drawn from the analysis of SIC and cloud radiative effect.

1. The PHACT dataset
   L43-44: "PHACT, which will be described in a separate paper".

   It is peculiar to read the analysis of a dataset without having the ability to read the details of the algorithm used to create the same dataset. In my opinion, the authors could describe the algorithm in more detail, in the absence of a peer-reviewed article supporting the use of the dataset in this publication, to equip the reader with the necessary tools to understand the reach and the limitations of the dataset.

   The diagnostics that are used in this paper are the same as the one described in Cesana and Chepfer (2013) and Cesana et al., (2016). A brief description of the CALIPSO-GOCCP dataset is given as well as the main differences between PHACT and GOCCP are mentioned in the dataset section. To make it more complete, we have added the description of the mixed-phase cloud category, which is not available in CALIPSO-GOCCP and that was missing in the previous version of the manuscript (L66-70): *"Additionally, PHACT contains a mixed-phase category which is not available in CALIPSO-GOCCP. Mixed-phase clouds are diagnosed when ice or undefined-phase clouds are retrieved below liquid, either directly underneath or not. Undefined-phase clouds are clouds that are located underneath highly reflective clouds (Cesana and Chepfer 2013}) and have been shown to be most likely mixed-phase clouds at subfreezing temperatures (Cesana et al., 2016)."*

   L45-46: "Cloud phase diagnostics are based on the cloud particle sphericity instead of temperature, in contrast with many passive sensors".
   It is possible to differentiate the phases not only by means of temperature profiles but also by differential complex refractive index at those wavelengths where ice and water absorption differ. What accuracy is expected with the proposed approach compared to others? What ranges of sphericity values are assigned to categorize the variety of droplets and crystals and everything in between

   We have compared CALIPSO-GOCCP to in situ aircraft observations and find that the maximum fraction difference for the phase, but it is actually 11.8 and 1% for two collocated in situ flights in the Arctic, when accounting for uncertainty in the in situ discrimination thresholds, which is mentioned in the description of PHACT (L71). For ice cirrus clouds, the cloud phase agreement fraction is close to 100% (Cesana et al., 2016).

2. Errors
   Within PHACT, the instantaneous profiles are aggregated at a spatial resolution of 2.5 degree. Recent studies (Kotarba, AR, 2022; Kotarba, AMT, 2022) have highlighted some shortcomings in the representation of climatological cloud fields by means of spaceborne lidars as function of grid resolution and cloud regimes. I think these results are relevant for the purposes of this article because it is clear that the errors introduced by undersampling (revisit and transect) can be considerable. I refer specifically to the figure 3 in the AMT article. The plots 3a-3c show the magnitude of cloud amount error with respect to general cloud regime (3a) and to sampling frequency (3c). I conclude that for high cloud amount CALIPSO sampling would result in absolute error of 5%, whereas for cloud amount of 60-80% the expected error would be 6-8% of cloud amount itself.

   Two recent papers noted these results (Winker et al, ESSD, 2023; Bertrand et al, ESSD, 2023)

and introduced correction factors to account for CALIPSO undersampling. It would be interesting to know the authors' opinion about this aspect and how this feature of CALIPSO impacts aggregate climatological cloud cover fields, differences with other datasets, and, finally, correlation values with sea ice extent.

This consideration brings me to ask what is the standard deviation of the curves in Figure 2 of this paper. The authors seem satisfied with the agreement among the datasets and it is certainly true that the seasonality is indeed in phase. But the absolute values are not in agreement and sometimes even exceeds the errors reported above in Kotarba AMT 2022. What if not even the standard deviation of the respective datasets overlap?

We do not expect perfect agreement between cloud covers among the datasets because the retrieval methods, the instruments, the cloud definitions, and the resolutions are different. Instead, we analyze consistency in the seasonality and the relationship with sea ice. Other studies have shown that CALIPSO retrieves cloud amount very well compared to other sensors (Cesana et al., 2019; Lacour et al., 2017; Protat et al., 2014; Stubenrauch et al., 2013).

Regarding the sampling uncertainty, the two Kobarta studies show that the effect is negligible when the data are averaged, which is what we're doing in our analysis. From what we understand of Winker et al. (2023) no correction is applied to their dataset, they only attempt to give an estimate of what could be the error related to this sampling issue on IWC. Their results show that at our latitudes, where the sampling is best, the error on IWC is negligible. We have computed differences in total, ice and liquid cloud covers between 1x1° grid and 2.5x2.5° grid for a year of data and we find no significant differences, as shown in the figure below. We have added this information at the end of the dataset section (L110-111): "*We note that we find no significant impact of the spatial resolution – i.e., using either 2.5˚x2.5˚ and 1˚x1˚ grid – for PHACT total, ice and liquid cloud covers over the Arctic (not shown).*"

[Figure]

3. Conclusions

3-a) The sentence at line 182-183 ("Such a cooling effect is found in all seasons but winter, when the LW CRE warming exceeds the SW CRE cooling.") seems inaccurate to me.

Arctic winter months are characterized by the polar night. Assuming the authors calculate the net CRE at the surface from the difference between the downward and upward flux components of SW and LW for all-sky and clear-sky conditions (is this how you compute CRE?), the absence of sunlight makes the terms SW_down and SW_up zero and only the LW emissive terms are present. Cloud cannot reflect shortwave radiation during the polar night, right?

The SW CRE is computed over the DJF season from 60 to 90°N. As shown in Fig. 8, there is some daylight until approximately 80°N. We have added "*defined as north of 60°N*" in the first sentence of the introduction (L17).

3-b) The last paragraph of the conclusion supports the results in Lelli et al. 2023.
A paper comes alive only if it ventures outside its own scientific project vacuum and it is related to previous research. Therefore, I think that the authors of this interesting manuscript could try to answer the following questions: (i) do your results contradict, confirm, strengthen previously established knowledge? (ii) which new research questions arise from the contrast between previous research and results here? (iii) How could further your research help resolve them? A better connection with past results is beneficial.

For instance, during the preparation of Lelli et al ACP 2023, we created the following figure where we look at CRE SW and LW above sea ice areas. The right panels show that the (negative) SW CRE trend increases for a decreasing sea ice extent, while the LW CRF trends slightly increase. Inspecting the crossing point between trends in SW and LW CRF as function of SIC, we see that already for a SIC change of 0.05%/month suffices for the SW CRF to offset the LW CRF. In these results the CRF changes as a function of cloud property changes are clearly conflated. Note that AMJ and JAS stand for Arctic spring (AMJ) and summer (JAS) and some differences might be due to the different season definition.

[Figure]

We additionally point the authors to Sections 3.2 onward in Lelli et al. 2023 for a detailed description of those cloud property changes and the CRE sensitivities. As a side note, cloud properties and fluxes have been extensively validated. The references of interest are given in Lelli et al. ACP 2023.
Our results are consistent with previous studies and brings additional insights as mentioned in our conclusions. In response to Reviewer 1, we have added trends of CERES flux anomalies along with sea-ice cover. Consistent with our sea ice vs. open ocean analysis, we find that as sea ice decreases the cooling effect of clouds in the SW strengthens while their warming effect in the LW remains neutral. We have added a reference to Lelli et al. 2023 in that section (L230-232): "*Overall, this strengthens the cooling effect of clouds at the surface (i.e., more negative SW CRE), which is consistent with previous literature looking at surface SW CRE trends during spring and summer (Lelli et al., 2023).*"

3-c) From what I can understand from the present manuscript, the authors analyzed CRE by relating it to the surface, treating it as if it were binary: no sea ice/sea ice. Where we know that in reality Arctic sea ice is a much more complex surface realm (leads, melt ponds, floes ... ) such that it likely results in

distribution of ice concentration (0-100%) within a 2.5 degree side-by-side grid cells of the chosen aggregation. Would it be possible for the authors to add a more refined dimension of analysis and treat Arctic sea ice (and related trends) as a continuum? How would the result change?

We do not treat sea ice as binary, we look at correlations between sea ice cover and cloud cover in regions where sea ice cover varies over the analyzed time period (L121-124 of the original manuscript). We somewhat treat sea ice cover as binary when we assume grid boxes to be dominated by sea ice or open ocean in Fig. 7 of the manuscript. Choosing different thresholds to restrict our analysis to grid boxes almost fully covered by sea ice or open water doesn't change our result qualitatively (see response to minor comment 2 of Reviewer 1).

3-d) I would like to ask whether the albedo of the analyzed sea ice surfaces can change the results found by the authors. CRE is by definition a net quantity, so it is to be related to the effectiveness of the surface to reflect SW. And this is a function of albedo, which in turn changes depending on whether - for example - there is fresh snow in the spring or old snow in the summer before the start of the melt season and so on. Can the authors comment on why they do not consider this aspect?

It is correct that the CRE would not be the same over different surface types for similar cloud properties. However, in our case, the CERES CRE is calculated assuming the same surface type on the all-sky and clear fluxes, so it doesn't affect our results. We have added this information in the description of the datasets (L94-96): "*The computation of all-sky and clear-sky fluxes assumes the same surface type, which means that the surface albedo is accounted for in the computations of cloud radiative effect (CRE) values (Section 3)*".

3-e) I think that is a bit of a missed opportunity - yet at authors' discretion - not to present also quantitative results on optical thickness of the clouds. In this way, the claims of this paper could be more substantiated and directly linkable to Lelli et al.

The main topic of this paper is to study the relationship between sea ice, cloud phase and radiation. To first order, the cloud phase is a good proxy of the cloud optical thickness. The increase in liquid clouds with decreasing sea ice cover is indicative of an increase of the overall optical thickness of clouds, which we mention in the conclusions. We also note that there are large uncertainties in liquid optical depth retrievals from MODIS (up to 80% in the polar regions, Steve Platnick, personal communication).

References

Cesana, G., Waliser, D. E., Henderson, D., L'Ecuyer, T. S., Jiang, X., & Li, J.-L. F. (2019). The Vertical Structure of Radiative Heating Rates: A Multimodel Evaluation Using A-Train Satellite Observations. *Journal of Climate*, *32*(5), 1573–1590. https://doi.org/10.1175/JCLI-D-17-0136.1

Lacour, A., Chepfer, H., Shupe, M. D., Miller, N. B., Noel, V., Kay, J., Turner, D. D., & Guzman, R. (2017). Greenland Clouds Observed in CALIPSO-GOCCP: Comparison with Ground-Based Summit Observations. *Journal of Climate*, *30*(15), 6065–6083. https://doi.org/10.1175/JCLI-D-16-0552.1

Protat, A., Young, S. A., McFarlane, S. A., L'Ecuyer, T., Mace, G. G., Comstock, J. M., Long, C. N., Berry, E., & Delanoë, J. (2014). Reconciling ground-based and space-based estimates of the frequency of occurrence and radiative effect of clouds around Darwin, Australia. *Journal of Applied Meteorology and Climatology*, *53*(2), 456–478. https://doi.org/10.1175/JAMC-D-13-072.1

Stubenrauch, C. J., Rossow, W. B., Kinne, S., Ackerman, S., Cesana, G., Chepfer, H., di Girolamo, L., Getzewich, B., Guignard, A., Heidinger, A., Maddux, B. C., Menzel, W. P., Minnis, P., Pearl, C., Platnick, S., Poulsen, C., Riedi, J., Sun-Mack, S., Walther, A., … Zhao, G. (2013). Assessment of global cloud datasets from satellites. *Bulletin of the American Meteorological Society*, *94*(7), 1031–1049. https://doi.org/10.1175/BAMS-D-12-00117.1